# CHAIN-OF-VERIFICATION REDUCES HALLUCINATION IN LARGE LANGUAGE MODELS

## ABSTRACT

Generation of plausible yet incorrect factual information, termed hallucination, is an unsolved issue in large language models. We study the ability of language models to deliberate on the responses they give in order to correct their mistakes. We develop the Chain-of-Verification (COVE) method whereby the model first (i) drafts an initial response; then (ii) plans verification questions to fact-check its draft; (iii) answers those questions independently so the answers are not biased by other responses; and (iv) generates its final verified response. In experiments, we show COVE decreases hallucinations across a variety of tasks, from list-based questions from Wikidata, closed book MultiSpanQA and longform text generation.

## 1 INTRODUCTION

Large Language Models (LLMs) are trained on huge corpora of text documents with billions of tokens of text. It has been shown that as the number of model parameters is increased, performance at tasks such as closed book QA improve in accuracy, and larger models can generate more correct factual statements (Radford et al., 2019; Petroni et al., 2019). However, even the largest models can still fail, particularly on lesser known torso and tail distribution facts (Sun et al., 2023a), i.e. those that occur relatively rarely in the training corpora. In those cases where the model is incorrect, they instead generate an alternative response which is typically plausible looking (e.g., a similar entity, but an incorrect one). These factually incorrect generations are referred to as hallucinations (Maynez et al., 2020). Further, in longform tasks consisting of generating multiple sentences or paragraphs, the hallucination problem can be exacerbated due to the issue of exposure bias (Wang & Sennrich, 2020).

The current wave of language modeling research goes beyond next word prediction, and has focused on their ability to reason. Improved performance in reasoning tasks can be gained by encouraging language models to first generate internal thoughts or reasoning chains before responding (Wei et al., 2022; Adolphs et al., 2021; Wang et al., 2022; Lanchantin et al., 2023), as well as updating their initial response through self-critique (Press et al., 2022; Madaan et al., 2023). In this work we follow this line of research to study how and when language-model-based reasoning can be used to reduce hallucinations. We develop an approach, called Chain-of-Verification (CoVe) which, given an initial draft response, first plans verification questions to check its work, and then systematically answers those questions in order to finally produce an improved revised response. We find that independent verification questions tend to provide more accurate facts than those in the original longform answer, and hence improve the correctness of the overall response. We study variations on this recipe across a range of tasks: from list-based questions, closed booked QA and longform text generation. We first propose a joint approach for generating the entire verification chain left-to-right, which improves performance and decreases hallucinations compared to the baseline language model. However, models that attend to existing hallucinations in the context from their own generations tend to repeat the hallucinations. Hence we also introduce further improvements with factored variants which separate out the verification chain steps, in terms of which context is attended to. We show how these factored variants give further performance gains across all three tasks considered.

## 2 RELATED WORK

Hallucination is a general problem in language model generations that appears across many tasks, from summarization (Maynez et al., 2020) to open-domain dialogue (Roller et al., 2020), and has not

**Query**

Name some politicians who were born in NY, New York.

**1. Baseline Response**

Here are some politicians who were born in NY, New York:
1. Hillary Clinton - former secretary of state and former Democratic presidential nominee
2. Donald Trump - former president of the United States
3. Michael Bloomberg - former Mayor of New York City and former Democratic presidential candidate
… <list continues..>

**2. Plan Verifications**

- Where was Hillary Clinton born?

- Where was Donald Trump born?

- Where was Michael Bloomberg born?

… <questions continue..>

**3. Execute Verifications**

Hillary Clinton was born in **Chicago, Illinois**, United States on October 26, 1947.

Donald Trump was born on June 14, 1946, in **Queens, New York City, New York**, United States

Michael Bloomberg was born on February 14, 1942, in **Boston, Massachusetts**, United States.

…

**4. Final Verified Response**

Here are some politicians who were born in NY, New York:
1. Donald Trump - former president of the United States
2. Alexandria Ocasio-Cortez - Democratic member of the U.S. House of Representatives
… <list continues..>

Figure 1: Chain-of-Verification (CoVe) method. Given a user query, a large language model generates a baseline response that may contain inaccuracies, e.g. factual hallucinations. We show a query here which failed for ChatGPT (see section 12 for more details). To improve this, CoVe first generates a plan of a set of verification questions to ask, and then executes that plan by answering them and hence checking for agreement. We find that individual verification questions are typically answered with higher accuracy than the original accuracy of the facts in the original longform generation. Finally, the revised response takes into account the verifications. The factored version of CoVe answers verification questions such that they cannot condition on the original response, avoiding repetition and improving performance.

been resolved by simply scaling up training data or model size (Zhang et al., 2023). For a survey of the hallucination issue, see Ji et al. (2023) and Ye et al. (2023). A majority of the methods for reducing hallucination can be divided into roughly three categories: training-time correction, generation-time correction and via augmentation (tool-use).

In training-time correction methods, an attempt is made to improve the raw left-to-right generations of an encoder-decoder or decoder-only language model by either training or otherwise adjusting the model weights to decrease the probability of hallucinated generations. This includes using reinforcement learning (Roit et al., 2023; Wu et al., 2023), constrastive learning (Chern et al., 2023b; Sun et al., 2023b) and other methods (Li et al., 2023; Narayan et al., 2023).

In generation-time correction, a common theme is to make reasoning decisions "on top of" the base LLM in order to make them more reliable. For example, by considering the probabilities of the generated tokens (Mielke et al., 2022; Kadavath et al., 2022). In Manakul et al. (2023) multiple samples are drawn from the model to detect hallucinations. In Varshney et al. (2023) hallucinations are identified using low confidence scores, and their correctness is checked through a validation procedure, mitigated, and then the generation is continued. An alternative to using the confidence scores is to leverage inconsistencies in the LLMs output to detect hallucination. Cohen et al. (2023) show that using inconsistencies for QA tasks can outperform using confidence scores for hallucination detection. Cohen et al. (2023) simulate an interactive multi-agent LM vs. LM debate setup to detect

hallucinations for factoid QA, Agrawal et al. (2023) check for hallucinated references, while Mündler et al. (2023) extract relational triples from generations and verify them against another LLM. CoVe also uses a related self-consistency approach but we show our approach can correct hallucinated facts in *longform* generations by generating and answering verification questions by *solely* using the same LLM.

A third approach is to use external tools to help mitigate hallucinations, rather than relying solely on the abilities of the language model itself. For example, retrieval-augmented generation can decrease hallucinations by using factual documents for grounding (Shuster et al., 2021; Jiang et al., 2023b; Yu et al., 2023) or chain-of-thought verification (Zhao et al., 2023). Other approaches include using tools for fact-checking (Chern et al., 2023a; Galitsky, 2023; Peng et al., 2023), or linking to external documents with attribution (Menick et al., 2022; Rashkin et al., 2023; Gao et al., 2023).

There are also a number of related works in improving reasoning for logical and mathematical tasks, even if they do not address reducing hallucination explicitly. Several approaches have been shown to improve results with extended reasoning steps by the system, such as chain-of-thought (Wei et al., 2022), deductive verification (Ling et al., 2023), and self-verification (Miao et al., 2023; Jiang et al., 2023a; Weng et al., 2022). The latter tries to predict the (masked) question given the answer for math problems, and use that as evidence that this is the correct solution.

## 3 CHAIN-OF-VERIFICATION

Our approach assumes access to a base LLM that – despite potentially being prone to hallucination – is capable of being prompted with general instructions in either a few-shot or zero-shot fashion. A **key assumption** of our method is that this language model, when suitably prompted, can both generate and execute a plan of how to verify itself in order to check its own work, and finally incorporate this analysis into an improved response.

Our overall process, which we call Chain-of-Verification (CoVe), thus performs four core steps:

1. *Generate Baseline Response*: Given a query, generate the response using the LLM.

2. *Plan Verifications*: Given both query and baseline response, generate a list of verification questions that could help to self-analyze if there are any mistakes in the original response.

3. *Execute Verifications*: Answer each verification question in turn, and hence check the answer against the original response to check for inconsistencies or mistakes.

4. *Generate Final Verified Response*: Given the discovered inconsistencies (if any), generate a revised response incorporating the verification results.

Each of these steps is performed by prompting the same LLM in different ways to obtain the desired response. While steps (1), (2) and (4) all can be invoked with a single prompt, we investigate variations of step (3) including joint, 2-step and factored versions. These variants either involve a single prompt, two prompts or else independent prompts per question, where more sophisticated decomposition can yield improved results.

We describe these steps in more detail below. An overview of the approach is illustrated in Figure 1, and in the Appendix in Figure 3.

### 3.1 BASELINE RESPONSE

Given a query, we generate left-to-right as usual using the LLM, with no special tricks. While this is the first step in the CoVe pipeline, it also serves as the baseline we wish to improve in our experiments (i.e., we will directly compare this baseline response with the final verified response from our overall method).

Given such baseline generations are typically prone to hallucination, CoVe attempts to identify these hallucinations, and correct them, in the following steps.

## 3.2 PLAN VERIFICATIONS

Conditioned on the original query and the baseline response, the model is prompted to generate a series of verification questions that test the factual claims in the original baseline response. For example if part of a longform model response contains the statement *"The Mexican–American War was an armed conflict between the United States and Mexico from 1846 to 1848"*, then one possible verification question to check those dates could be *"When did the Mexican American war start and end?"*. We note that verification questions are not templated and the language model is free to phrase these in any form it wants, and they also do not have to closely match the phrasing of the original text.

In our experiments, we perform such verification planning by providing a few-shot prompt of (response, verification) demonstrations to our LLM. See section 11 for the few-shot prompts we will use in our experiments. We note it is also possible with a sufficiently performant instruction-following LLM that this could be performed zero-shot.

## 3.3 EXECUTE VERIFICATIONS

Given the planned verification questions, the next step is to answer them in order to assess if any hallucinations exist. While techniques such as retrieval-augmentation could be used in this process, such as verification via search engine, in this work we do not explore tool-use. Instead, we consider only using the LLM itself in all steps of CoVe, hence the model is used to check its own work. We investigate several variants of verification execution, called joint, 2-Step, factored and factor+revise.

**Joint**   In the *joint* method, the planning and execution (steps 2 and 3) are accomplished by using a single LLM prompt, whereby the few-shot demonstrations include both verification questions and their answers immediately after the questions. In this approach separate prompts are not needed.

**2-Step**   A potential disadvantage of the *joint* method is that because the verification questions must condition on the baseline response in the LLM context, and the method is joint, the verification answers have to condition on the initial response as well. This may increase the likelihood of repetition, another known issue of modern LLMs (Holtzman et al., 2019). This means the verification questions might hallucinate similarly to the original baseline response, which defeats the purpose. We hence instead separate the planning and execution into separate steps, both with their own LLM prompt. The planning prompt conditions on the baseline response in the first step. The verification questions generated from planning are answered in the second step, where crucially the context given to the LLM prompt only contains the questions, and not the original baseline response and hence cannot repeat those answers directly.

**Factored**   Another, more sophisticated approach, is to answer all questions independently as separate prompts. Again, crucially, those prompts do not contain the original baseline response and are hence not prone to simply copying or repeating it. The factored approach has the further advantage of removing any potential interference not only from the baseline response, but also between answer contexts, and is somewhat related to the recent (concurrent) work of Radhakrishnan et al. (2023) for subquestion answering by factored decomposition, hence we adopt their naming. It can also potentially handle more verification questions by virtue of them not all having to fit with the same single context. While this is potentially more computationally expensive, requiring the execution of many more LLM prompts, they can be run in parallel, and hence be batched. In order to do this, we first have to take the set of generated questions from subsection 3.2 and parse them into separate questions, which is a relatively easy task as the few-shot demonstrations we provide indicate they should be generated as a comma-separated list. We can then split them out into separate LLM prompts.

**Factor+Revise**   After answering the verification questions, the overall CoVe pipeline then has to either implicitly or explicitly cross-check whether those answers indicate an inconsistency with the original responses. In the factor+revise approach, we execute this as a deliberate step via an extra LLM prompt, which may make it easier for the final system to reason about this step explicitly. Differently to answering the verification questions, the cross-checking phase needs to condition on both the baseline response and the verification question and answer. We thus execute this as separate LLM prompts, one "cross-check" prompt for each question, with again a set of few-shot

demonstrations showing the desired output. For example if the original baseline response contained the phrase *"It followed in the wake of the 1845 U.S. annexation of Texas. . . "* and CoVe generated a verification question *When did Texas secede from Mexico?* which was answered with *1836* then an inconsistency should be detected by this step.

### 3.4 FINAL VERIFIED RESPONSE

Finally, the improved response that takes verification into account is generated. This is executed by a final few-shot prompt where the context takes into account all of the previous reasoning steps, the baseline response and verification question answer pairs, so that the corrections can take place. If the Factor+Revise approach is used from subsection 3.3 then the output of the cross-check inconsistency detection is provided as well.

## 4 EXPERIMENTS

We use various experimental benchmarks to measure the efficacy of CoVe in reducing hallucination, comparing against a number of baselines.

### 4.1 TASKS

The benchmarks we use range from list-based questions where the required answer is a set of entities, to where the answer is a longform generation of multiple freeform sentences.

#### 4.1.1 WIKIDATA

We start by testing CoVe on a set of automatically generated questions using the Wikidata API[1]. We create list questions of the form: "Who are some [Profession]s who were born in [City]?". For example, "Who are some politicians who were born in Boston?". The answer to these questions is a set of entities, where the gold list is obtained from the Wikidata knowledge base. This results in a dataset of 56 test questions, each typically containing ∼600 known gold entities, but typically an LLM will produce a much shorter list. We then use the precision metric (micro-averaged) to measure performance, in addition to reporting the averaged number of positive and negative entities produced.

#### 4.1.2 WIKI-CATEGORY LIST

We then proceed to a harder set-generation task. We use the QUEST (Malaviya et al., 2023) dataset that was created using Wikipedia Category lists. We convert these category names to questions by simply prepending a "Name some". Owing to the varied questions such as *Name some Mexican animated horror films* or *Name some Endemic orchids of Vietnam* we believe this task can pose a greater challenge. We collate all examples in the dataset that *do not require* logical operations to create a set of 55 test questions each having ˜8 answers. Similar to the Wikidata task, we measure precision (micro-averaged) to measure performance, in addition to reporting the averaged number of positive and negative entities produced.

#### 4.1.3 MULTISPANQA

We next test our approach on an reading comprehension benchmark, MultiSpanQA (Li et al., 2022). MultiSpanQA comprises of questions that have multiple independent answers (derived from a series of multiple discontiguous spans in the text, with questions originally from the Natural Questions dataset). We consider a closed-book setting, where we do not provide supporting documents, and hence consider a subset of questions which are factoid-based, so that our base LLM is more likely to be able to answer them. We thus use a test set of 418 questions with shorter answers per span (up to 3 tokens per item). For example, Q: Who invented the first printing press and in what year?, A: *Johannes Gutenberg, 1450*.

---

[1]https://query.wikidata.org/

Table 1: Test Precision and average number of positive and negative (hallucination) entities for list-based questions on the Wikidata and Wiki-Category list tasks.

| | | Wikidata (Easier) | | | Wiki-Category list (Harder) | | |
| --- | --- | --- | --- | --- | --- | --- | --- |
| LLM | Method | Prec. ($\uparrow$) | Pos. | Neg. | Prec. ($\uparrow$) | Pos. | Neg. |
| Llama 2 70B Chat | Zero-shot | 0.12 | 0.55 | 3.93 | 0.05 | 0.35 | 6.85 |
| Llama 2 70B Chat | CoT | 0.08 | 0.75 | 8.92 | 0.03 | 0.30 | 11.1 |
| Llama 65B | Few-shot | 0.17 | 0.59 | 2.95 | 0.12 | 0.55 | 4.05 |
| Llama 65B | CoVe (joint) | 0.29 | 0.41 | 0.98 | 0.15 | 0.30 | 1.69 |
| Llama 65B | CoVe (two-step) | **0.36** | 0.38 | 0.68 | 0.21 | 0.50 | 0.52 |
| Llama 65B | CoVe (factored) | 0.32 | 0.38 | 0.79 | **0.22** | 0.52 | 1.52 |

### 4.1.4 LONGFORM GENERATION OF BIOGRAPHIES

We next validate the performance of CoVe on longform text generation. In this setting, we evaluate our method on generating biographies, adopting the benchmark proposed in by Min et al. (2023). Here the model is simply prompted to generate a biography of a selected entity using the prompt: "Tell me a bio of <entity>". We evaluate the efficacy of our approach using the FACTSCORE metric (Min et al., 2023) developed in that work, which uses a retrieval-augmented language model to fact-check the response (Instruct-Llama, "Llama + Retrieval + NP"), which they showed correlates well with human judgments.

### 4.2 BASELINES

We use Llama 65B, a strong open model as our base LLM (Touvron et al., 2023a), and use greedy decoding for all models. As Llama 65B is not instruction fine-tuned, we employ few-shot examples particular to each task for measuring performance on each of our benchmarks. This serves as our main baseline which CoVe tries to improve upon. CoVe uses the same Llama 65B base, but includes, for the same few-shot examples, demonstrations of verification questions and final verified responses, following Figure 1 and section 3. Thus, we measure the ability to improve over the original baseline response for the same LLM. For CoVe, we compare different variants, particularly the joint and factored versions on all tasks.

We also compare to Llama instruction fine-tuned models, for which we use Llama 2 (Touvron et al., 2023b). We measure both zero-shot performance on the task, or zero-shot with chain-of-thought by adding "Let's think step by step" to the zero-shot prompt. We find that the instruction fine-tuned models tend to generate extraneous content when queried. This can especially be a problem for the list-based tasks. To deal with this we add an extra line to our prompt: "List only the answers separated by a comma". We also add another layer of post-processing to extract the answers by using an off-the-shelf NER model to further avoid this issue as this helped. However, we still expect few-shot to improve over this, especially for tasks like Multi-Span-QA where the answers are not all named entities, and the few-shot examples effectively show the domain of the task.

For the longform generation of biographies we also compare to several existing model results reported in Min et al. (2023), in particular InstructGPT (Ouyang et al., 2022), ChatGPT [2] and PerplexityAI [3].

### 4.3 RESULTS

We are interested in empirically answering the following research questions:

**RQ1** Can COVE effectively reduce the rate of hallucinatory content produced by the LLM?

**RQ2** Can COVE be used to fix or remove incorrect generations without decreasing the amount of correct content?

---

[2] https://openai.com/blog/chatgpt
[3] www.perplexity.ai

Table 2: Closed book MultiSpanQA test performance, comparing CoVe with various baselines.

| LLM | Method | F1 (↑) | Prec. | Rec. |
|---|---|---|---|---|
| Llama 2 70B Chat | Zero-shot | 0.20 | 0.13 | 0.40 |
| Llama 2 70B Chat | CoT | 0.17 | 0.11 | 0.37 |
| Llama 65B | Few-shot | 0.39 | 0.40 | 0.38 |
| Llama 65B | CoVe (joint) | 0.46 | 0.50 | 0.42 |
| Llama 65B | CoVe (factored) | **0.48** | 0.50 | 0.46 |

Table 3: Longform generation of biographies with metrics defined from Min et al. (2023). Models marked with ∗ are reported from previous work. FACTSCORE automatically computed using "Instruct-Llama" ( Retrieve → LM + NP), the best open-access model.

| LLM | Method | FACTSCORE. (↑) | Avg. # facts |
|---|---|---|---|
| InstructGPT* | Zero-shot | 41.1 | 26.3 |
| ChatGPT* | Zero-shot | 58.7 | 34.7 |
| PerplexityAI* | Retrieval-based | 61.6 | 40.8 |
| Llama 2 70B Chat | Zero-shot | 41.3 | 64.9 |
| Llama 2 70B Chat | CoT | 41.1 | 49.0 |
| Llama 65B | Few-shot | 55.9 | 16.6 |
| Llama 65B | CoVe (joint) | 60.8 | 12.8 |
| Llama 65B | CoVe (factored) | 63.7 | 11.7 |
| Llama 65B | CoVe (factor+revise) | **71.4** | 12.3 |
| GPT-3 | Few-shot | 45.3 | 15.6 |
| GPT-3 + ChatGPT | ChatProtect Mündler et al. (2023) | 48.5 | 14.6 |
| GPT-3 + InstructGPT | SCG-LL Manakul et al. (2023) | 60.6 | 6.0 |
| GPT-3 + DeBERTA | SCG-NLI Manakul et al. (2023) | 61.7 | 6.3 |
| GPT-3 + InstructGPT | CoVe (factor+revise) | **68.6** | 9.0 |

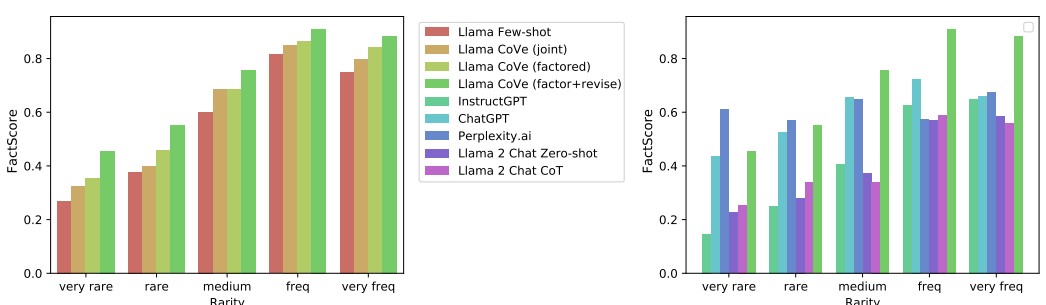

Figure 2: FACTSCORE performance distribution across head, torso and tail facts for CoVe variants and various baselines on longform generation of biographies.

Our main results across the four benchmark tasks are given in Table 1, Table 2 and Table 3, and our main findings are as follows.

**CoVe improves precision on list-based answer tasks** We find that CoVe provides large gains in precision on the list-based tasks, e.g. more than doubles the precision from the Llama 65B few-shot baseline for the Wikidata task (from 0.17 to 0.36). We find from the positive and negative breakdown that there is a large reduction in the number of hallucinated answers (negatives: 2.95 → 0.68) while only a relatively small reduction in the number of non-hallucinations (positives: 0.59 → 0.38).

**CoVe improves performance on closed book QA**    We also find that CoVe brings improvements in general QA problems, as measured on MultiSpanQA. We observe a 23% improvement in F1 over the few-shot baseline ($0.39 \rightarrow 0.48$), where the improvements come from gains in both precision and recall.

**CoVe improves precision on longform generation**    These results also extend to longform generation, where we actually see larger gains than in the QA setting. FACTSCORE increases 28% ($55.9 \rightarrow 71.4$) from the few-shot baseline, with again only a relatively small reduction in average number of facts provided ($16.6 \rightarrow 12.3$). We also show the breakdown of improvements across facts in Figure 2, where one can see CoVe improves results for both rare and more frequent facts.

**Instruction-tuning and CoT do not reduce hallucinations**    We find that the few-shot baseline that employs a pre-trained Llama model outperforms Llama-2-Chat, an instruction tuned model, across all the tasks. The few-shot examples lead the model to give outputs in line with those expected for the task, whereas general instruction tuning produces more hallucinations or incorrect outputs. Zero-shot chain-of-thought (CoT) prompting also fails to improve the results for these tasks.

**Factored and 2-step CoVe improve performance**    We observe a consistent performance improvement across all tasks from applying the factored CoVe approach compared to joint CoVe. For example improvement from $60.8 \rightarrow 63.7$ in FACTSCORE in longform generation. Similarly, the 2-step approach also outperforms the joint approach, as tested on the Wikidata and Wiki-Category list tasks, with 2-step giving the best results for Wikidata, and factored the best for Wiki-Category. All these results support our hypothesis that verifying questions should not attend to the original baseline response as they may be prone to repeating it (as the joint method can do).

**Further explicit reasoning helps remove hallucinations**    In the longform generation task we also explore more sophisticated reasoning steps in the CoVe "factor+revise" method, which explicitly cross-checks whether verification answers indicate an inconsistency. We see large gains in the FACTSCORE metric from this further explicit reasoning from 63.7 (factored) $\rightarrow$ 71.4 (factor+revise). This gives further indication that appropriate and explicit reasoning in LLMs can bring improvements in mitigating hallucinations.

**CoVe-based Llama outperforms InstructGPT, ChatGPT and PerplexityAI**    On the longform generation task, our baseline few-shot Llama 65B is outperformed by the ChatGPT and PerplexityAI models in terms of the FACTSCORE metric. However, applying CoVe to Llama 65B lifts its performance above ChatGPT and InstructGPT, as well as PerplexityAI. This is particularly impressive compared to PerplexityAI considering that is a model that can support its facts with retrieval-augmentation, whereas CoVe uses only the base language model itself with improved reasoning via deliberation (verification). However, Figure 2 shows PerplexityAI still outperforms CoVe for very rare facts where retrieval is essential, but CoVe outperforms PerplexityAI for frequent facts.

**CoVe outperforms existing hallucination mitigation baselines**    We compare CoVe to three recently released longform hallucination mitigation approaches — NLI and LLM, the two best-performing variants of SelfCheckGPT (Manakul et al., 2023) and ChatProtect Mündler et al. (2023). Our results in Table 3 show that COVE outperforms all these baselines. More details on our implementations of the baselines can be found in section 9. For SelfCheckGPT we experiment with different thresholds (in Table 6) and choose a threshold that results in a high FACTSCORE without removing a lot of facts. We note that some models produce fewer overall facts than others, however, the FACTSCORE metric is normalized and hence comparable across models. We verified this experimentally by clipping Llama-2-70B chat's output to present fewer facts (as it contains the largest number in its output out of all models), but this did not change its FACTSCORE substantially, e.g. clipping to 10 sentences increased its score from $41.3 \rightarrow 42.7$. We note the length of the generations of the few-shot-based models is essentially governed by the few-shot examples, which in turn are constrained by the context length.

**Shortform verification questions are more accurately answered than longform queries**    In a longform response, LLMs are prone to generate a number of hallucinations. However, it can often be the case that the LLM itself would know these hallucinations are wrong if queried specifically for

that individual fact, independent of the rest of the longform generation, see Figure 1, Figure 3, and section 12. This can be seen quantitatively on the Wikidata task, where only ∼17% of the Llama few-shot baseline answer entities are correct in list-based questions. However, when querying each individual entity via a verification question, we find ∼70% are correctly answered.

**Open LLM-based verification questions outperform yes/no-based and rule-based verification questions** In our method, CoVe, the verification questions generated by the LLM expect answers that are true facts. Another alternative type of verification questions would be templated verification questions, which can be generated cheaply, and binary questions. We first compare to heuristically constructed questions by replacing the LLM questions with templated yes/no questions of the form "Does $X$ answer the question" for list-based questions with elements $X$ in the answer. Results on the Wiki-Category task, given in Table 4, show a reduced precision with rule-based verification questions. We then move on to binary questions where we include the fact as part of the verification question and ask it in a yes/no answer format. We evaluate this difference in Table 4, and find that yes/no type questions perform worse for the factored version of CoVe. Some anecdotal examples are included in Appendix section 12 for ChatGPT where we find the model tends to agree with facts in a yes/no question format whether they are right or wrong. We believe this difference would be larger for longform generation where the types of required verification questions can be more diverse, and LLM-based verification becomes even more necesary.

**Inference overhead** Approaches that detect hallucinations via inconsistencies require repeated prompts to the LLM. CoVE essentially requires 1 LLM call for generating the baseline response, 1 LLM call for each sentence to plan the verifications, 1 LLM call to verify each fact and 1 LLM call for each fact to generate the consistent response. We also note that the LLM calls for verification for each fact can be parallelized. This is comparable and in certain cases fewer LLM calls compared to other approaches for hallucination mitigation. We provide a more in-depth analysis in section 8.

## 5 CONCLUSION

We introduced Chain-of-Verification (CoVe), an approach to reduce hallucinations in a large language model by deliberating on its own responses and self-correcting them. In particular, we showed that models are able to answer verification questions with higher accuracy than when answering the original query by breaking down the verification into a set of simpler questions. Secondly, when answering the set of verification questions, we showed that controlling the attention of the model so that it cannot attend to its previous answers (factored CoVe) helps alleviate copying the same hallucinations. Overall, our method provides substantial performance gains over the original language model response just by asking the same model to deliberate on (verify) its answer. An obvious extension to our work is to equip CoVe with tool-use, e.g., to use retrieval augmentation in the verification execution step which would likely bring further gains.

## 6 LIMITATIONS

While our Chain-of-Verification (CoVe) method seeks to reduce hallucinations, it does not remove them completely from generations. This means that CoVe can still generate incorrect or misleading information for a given query, even if it improves over the baseline. We also note that in our experiments we have only addressed hallucinations in the form of directly stated factual inaccuracies. However, hallucinations could come in other forms, such as during incorrect reasoning steps, as part of opinions, etc. We also note that the generations CoVe produces come with verifications which, if viewed by the user, add more interpretability to its decisions, but come at the cost of increased computational expense due to generating more tokens in the output, similar to other reasoning methods such as Chain-of-Thought.

Our method seeks to make a large language model produce improved responses by spending more time deliberating to identify its own mistakes. While we have shown this gives clear improvements, the upper bound to the improvement is clearly limited by the overall capabilities of the model, e.g. in identifying and knowing what it knows. In this regard, an orthogonal line of research, as discussed in section 2 is the use of external tools by language models, to gain further information beyond what is stored in its weights. While we do not explore that avenue in this work those techniques would likely be fruitful to combine with the findings here.

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

## 7 RULE-BASED VS BINARY VS GENERAL QUESTIONS

Table 4: Comparison of various CoVE verification plan strategies (rows) and verification execution techniques (columns) on the WikiCategory task.

|  | Verification Execution | |
|  | CoVe (joint) | CoVe (factored) |
| Verification Plan | Prec. | Prec. |
| Rule-based questions | 0.13 | 0.16 |
| *Generated by model:* | | |
| yes/no questions | 0.15 | 0.19 |
| general questions | 0.15 | 0.22 |

## 8 INFERENCE OVERHEAD

$N :=$ Number of text samples, 4 in our experiments

$k :=$ Number of repeated mitigations, 3 in our experiments

$s :=$ Number of sentences in the generated passage

$f :=$ Number of facts in the generated passage

| Method | # LLM prompts |
| --- | --- |
| Few-shot | 1 |
| ChatProtect Mündler et al. (2023) | $1 + s \times k \times (3)$ |
| SCG-LLM Manakul et al. (2023) | $n + 1 + s \times n$ |
| CoVe (factor+revise) | $1 + s + 2 \times f$ |

Table 5: Worst-case LLM prompts used by each hallucination mitigation approach in the worst case

Based on the current implementations we find the CoVE induces a similar inference overhead to other longform hallucination mitigation approaches.

## 9 IMPLEMENTATION OF BASELINES

We use the existing models provided by Manakul et al. (2023) for the NLI model. For the LLM model we prompt `GPT-3.5-turbo-instruction` with the following instruction (as described in Manakul et al. (2023).

```
Context: {}
Sentence: {}
Is the sentence supported by the context above?
Answer Yes or No:
```

| | NLI | | LLM | |
| Threshold | FS | Facts | FS | Facts |
| --- | --- | --- | --- | --- |
| High | 76.5 | 3.0 | 61.7 | 6.3 |
| Med | 60.6 | 6.0 | 56.8 | 8.4 |
| Low | 53.0 | 8.0 | 53.0 | 10 |

Table 6: Different threshold results for SelfCheckGPT

## 10 CoVe - Further details

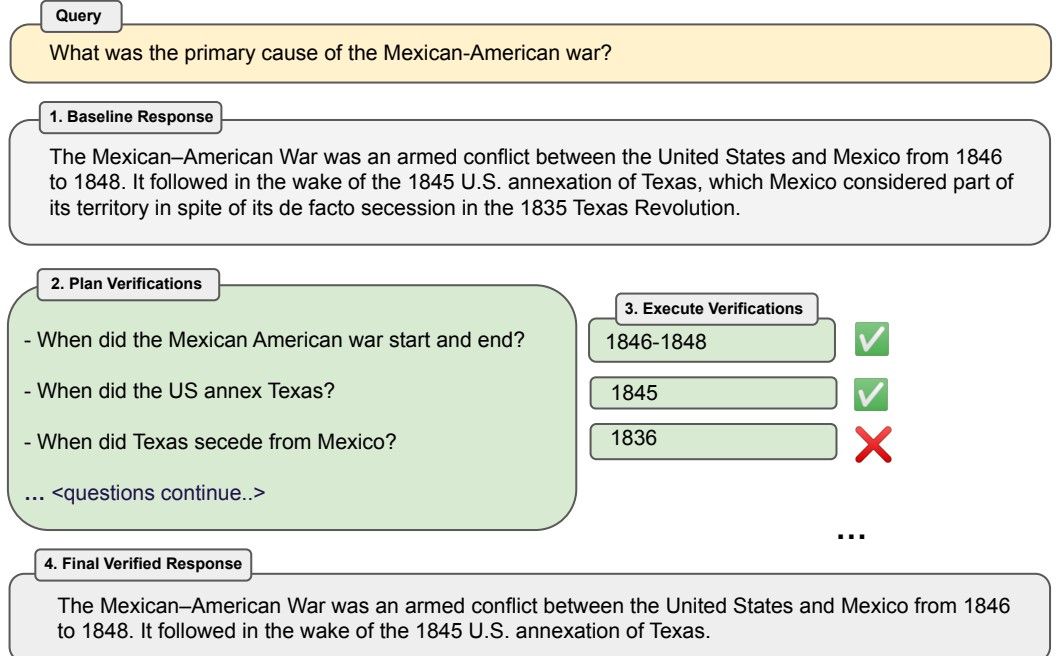

Figure 3: For longform generation, the Chain-of-Verification (CoVe) Factor + Revise method is the most effective in our longform generation experiments. CoVe Factor + Revise has the model independently identify (cross-check) which facts are consistent with its executed verifications (indicated by tickmark and crosses in the figure). With this extra step we aim to disregard the inconsistent facts and use the consistent facts to regenerate the response.

## 11    PROMPT TEMPLATES

We provide prompt templates for the longform generation of biographies task below for the different steps and variants of CoVe (see section 3). Templates for the other tasks are similar, but using few-shot examples from those tasks instead.

### 11.1    GENERATE BASELINE RESPONSE

```
Q: Tell me a bio of <person>
A: <bio of person>

Q: Tell me a bio of <person>
A: <bio of person>

Q: Tell me a bio of <person>
A: <bio of person>

Q: Tell me a bio of <person>
A:
```

Table 7: Few-shot prompting with 3 few-shot examples for the longform generation of biographies task. Other tasks use the same standard few-shot setup as well (with 3 examples from that particular task).

### 11.2    PLAN VERIFICATIONS

```
Context: Q: Tell me a bio of <person>.
A: <passage about person>
Response:
<fact in passage>, Verification Question
<fact in passage>, Verification Question

Context: Q: Tell me a bio of <person>.
A: <passage about person>
Response:
<fact in passage>, Verification Question
<fact in passage>, Verification Question

Context: Q: Tell me a bio of <person>.
A: <passage about person>
Response:
<fact in passage>, Verification Question
<fact in passage>, Verification Question

Context: Q: Tell me a bio of <person>.
A: <passage about person>
Response:
```

Table 8: Step (2) of CoVe involves planning the verification questions. In the biography task case we split the longform generation into its individual passages (e.g. sentences in the biography case, this was done due to excessive context length, which we don't need to do for the other tasks). The model then generates a verification question for each fact it observes in each passage (a passage may have multiple facts).

## 11.3 EXECUTE VERIFICATIONS

```
Q: Verification Question
A: Answer

Q: Verification Question
A: Answer

Q: Verification Question
A: Answer

Q: Verification Question
A:
```

Table 9: In step (3) of CoVe, the model then generates an answer for each of the verification questions. Again we use 3 few-shot examples.

## 11.4 GENERATE FINAL VERIFIED RESPONSE

```
Context: <Original Passage>.
From another source,
<output of execute verification step: Q + A>
<output of execute verification step: Q + A>
Response: <revised and consistent Passage>

Context: <Original Passage>.
From another source,
<output of execute verification step: Q + A>
<output of execute verification step: Q + A>
Response: <revised and consistent Passage>

Context: <Original Passage>.
From another source,
<output of execute verification step: Q + A>
<output of execute verification step: Q + A>
Response: <revised and consistent Passage>

Context: <Original passage>.
From another source,
<output of execute verification step: Q + A>
Response:
```

Table 10: In step (4) of CoVe (factored) the model is then presented with its original generation (split into passages, e.g. sentences, in the biography case, due to excessive context length which we do not need to do for the other tasks) along with its own verification step results. The model is told that this information comes from "another source". The model is required to synthesize a new final answer based on facts that are consistent between the two sources.

## 11.5 FACTOR+REVISE: IDENTIFY WHICH FACTS ARE CONSISTENT

```
Context: <Original Fact>.
From another source,
<output of execute verification step: Q + A>
Response: CONSISTENT. <Consistent fact>

Context: <Original Fact>.
From another source,
<output of execute verification step: Q + A>
Response: INCONSISTENT.

Context: <Original Fact>.
From another source,
<output of execute verification step: Q + A>
Response: PARTIALLY CONSISTENT. <Consistent part>
```

Table 11: In the CoVe (Factor + Revise) variant, as part of step (3) after subsection 11.3, the model is made to explicitly identify which facts are consistent between the two sources. The consistent facts can then be spliced together.

## 12    CHATGPT EXAMPLE SCREENSHOTS

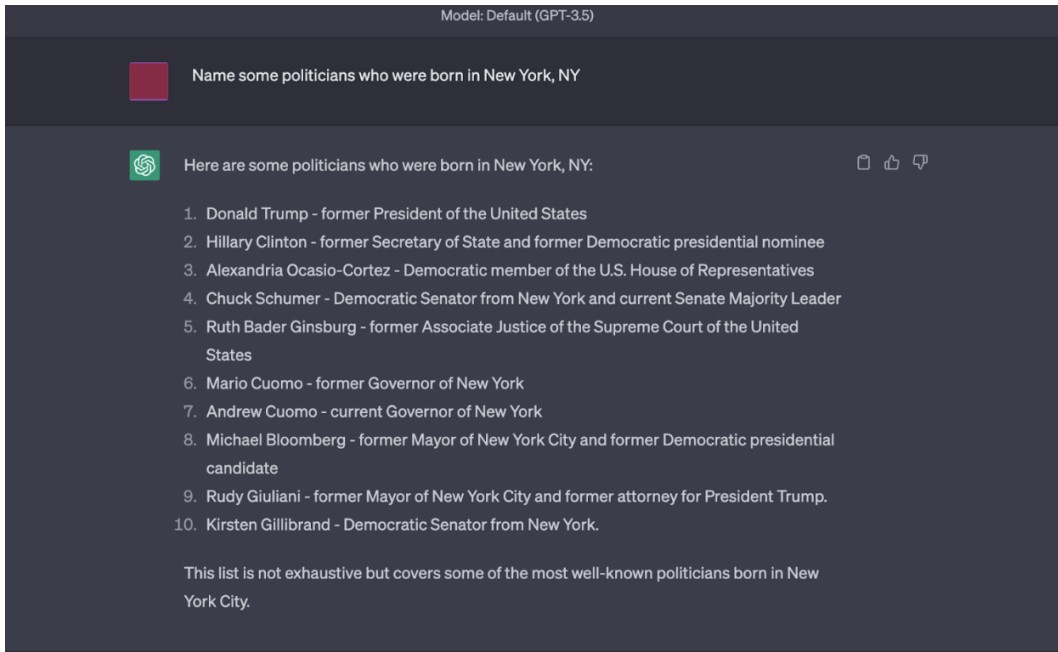

Figure 4:   ChatGPT generates several hallucinations for this question, e.g. Hillary Clinton and Michael Bloomberg.

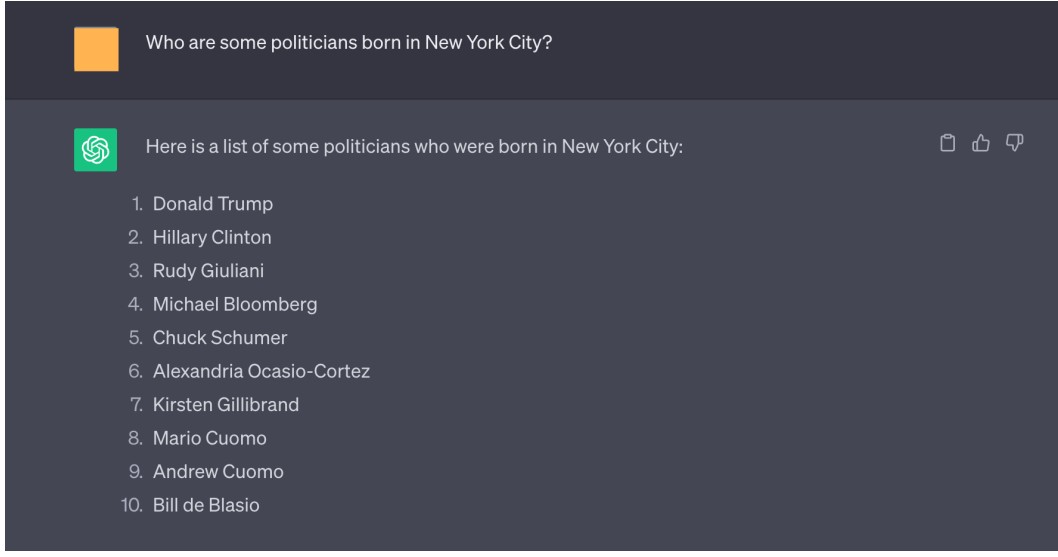

Figure 5:   Even when the longform answer is provided for a rewritten query (see query from Figure 4), while giving a slightly different answer, ChatGPT still generates several hallucinations for this question, e.g. Hillary Clinton and Michael Bloomberg.

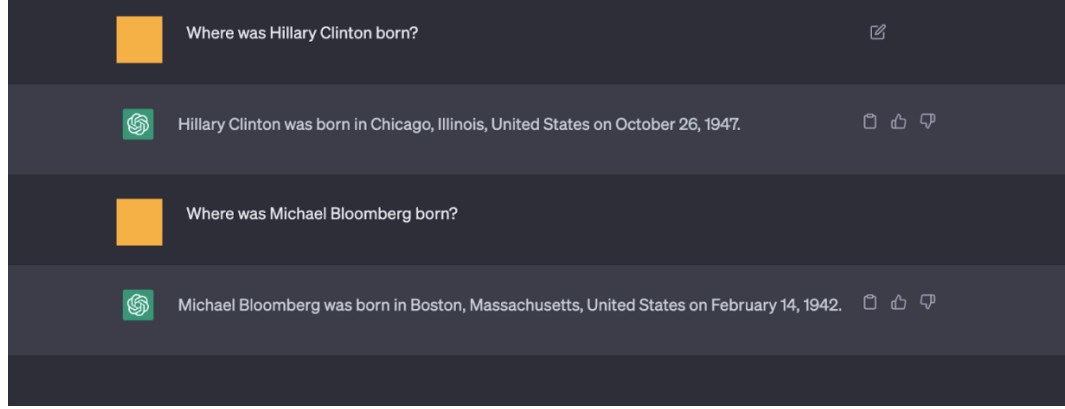

Figure 6: Shortform questions (which could be verification questions) appear to be answered more factually than the longform answers in Figure 4 and Figure 5.

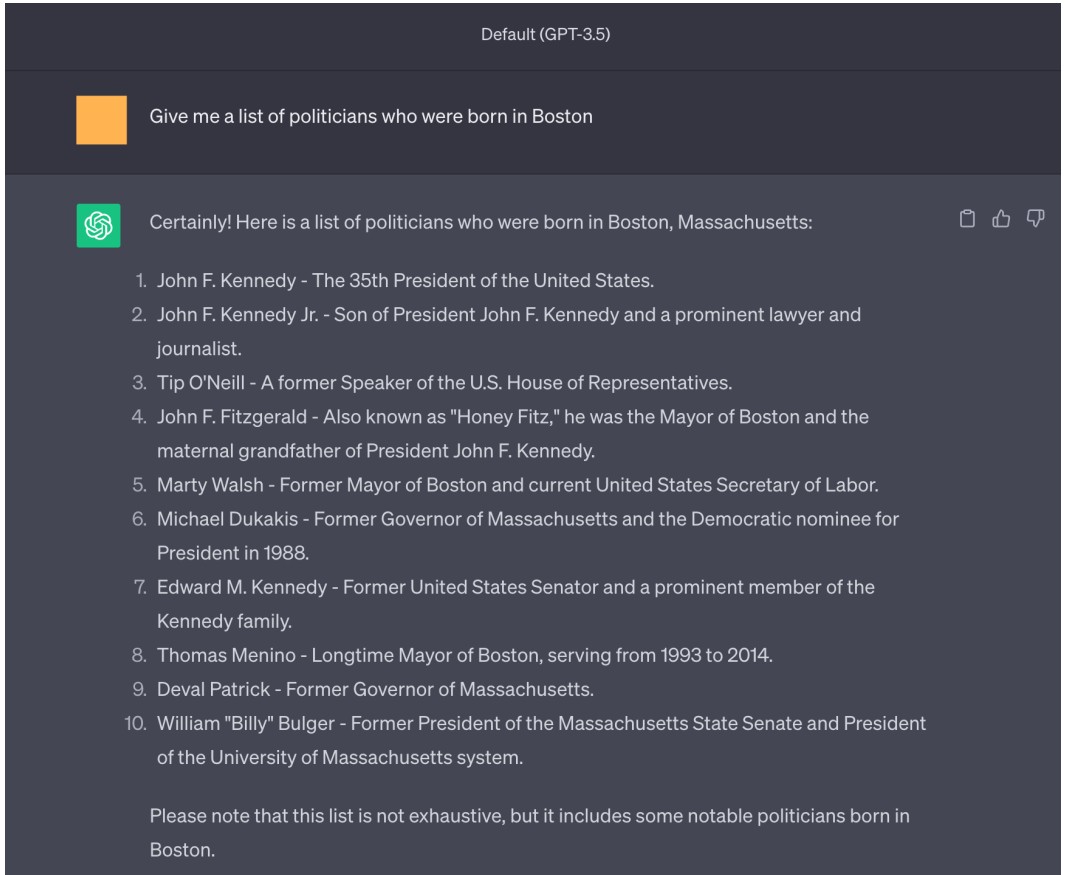

Figure 7: Another example of hallucinations for a different query, e.g., John F. Kennedy Jr was born in Washington D.C.

Figure 8: Examples where questions asking for a fact are answered correctly, but verifying via a yes/no question is incorrect (the model tends to agree with the way the question is stated, even if it was stated incorrectly).

