# OpenReview forum: "Chain-of-Verification Reduces Hallucination in Large Language Models"
_ICLR.cc/2024/Conference — Submitted to ICLR 2024_

### Official Review · Reviewer_gDTz · 2023-10-28

**Soundness:** 3 good
**Presentation:** 3 good
**Contribution:** 2 fair
**Rating:** 6
**Confidence:** 4

**Summary:**

This study introduces an approach aimed at mitigating hallucinations by harnessing the self-verification capabilities of Large Language Models (LLMs). The proposed method involves generating verification questions by an LLM to cross-check the accuracy of its initial responses, autonomously providing answers to these queries, and ultimately generating a refined response. The authors conducted a series of experiments to empirically establish the efficacy of this approach in addressing hallucination problems exhibited by the LLM across a range of tasks.

**Strengths:**

(1) The Chain-of-Verification concept is straightforward and can be practically implemented without necessitating adjustments to LLM's parameters.

(2) The empirical evaluations conducted on a varity of tasks, including list-based questions (Wikidata), closed-book MultiSpanQA, and longform text generation, demonstrate the effectiveness of the proposed method in mitigating LLMs' hallucinations.

(3) This paper is well-written and presents its ideas in a clear and comprehensible manner.

**Weaknesses:**

(1) The introduction of the proposed method does incur additional inference overhead. It would enhance the paper's rigor to compare these added computational costs with those associated with alternative methods that also target the reduction of LLM hallucinations.

(2) The utilization of few-shot learning for enabling LLMs to perform planning, verification, and generation (3-shot as detailed in the Appendix) raises a potential concern that the results might be influenced by variations in the few-shot examples used.

**Questions:**

(1) What criteria were employed for the selection of few-shot examples, and what is the potential influence of employing different examples on the results?

(2) Could you provide an estimate of the additional computational overhead that the proposed method would introduce?

---

> ### Author Response · Authors · 2023-11-23
>
> Thanks for the review and the positive comments on our approach’s ease of implementation, on the breadth of our experimentation, and the clarity of our draft!
>
> We have responded to your concerns below.
>
> **Inference overhead**
>
> Please check the common response. We also provide further comparative analysis in section 8 in our appendix.
>
>
> **Few-shot examples**
>
> We have now added the few-shot prompts used in our supplementary material for reproducibility.  We agree that changing the few-shot prompts can lead to a variance in the final output. In our comparative analysis, we use the same prompts to generate the baseline response for all our baseline approaches.
>
>
>
> **Questions:**
>
> 1. Few-shot prompts:
>       1. Our few-shot prompts for the list-based QA benchmarks were randomly chosen from the train set. We however keep these
>                   prompts constant across our variants.
>       2. For the biographies dataset, we choose some of the most viewed people's pages on Wikipedia and craft our prompts
>                   based on their biographies.  We find few-shot examples that allow us to show the model our desired verification responses.
>                   We have added the few-shot prompts used to the supplementary material.
> 2. We perform an analysis of the inference overhead. Please see the common response and Section 8 in our appendix.

---

### Official Review · Reviewer_kMJo · 2023-10-31

**Soundness:** 3 good
**Presentation:** 3 good
**Contribution:** 2 fair
**Rating:** 5
**Confidence:** 3

**Summary:**

Large Language Models (LLMs) can sometimes exhibit "hallucination," which refers to the generation of factually incorrect or misleading information. This work proposes the Chain-of-Verification (CoVe) method, a strategy for self-correcting LLM responses by asking and answering verification questions. The experimental results demonstrate that the CoVe approach reduces hallucinations in a variety of tasks, including Wikidata-based list problems, closed-book MultiSpanQA, and long-form generation.

**Strengths:**

1) The writing is well.
2) The idea of correcting LLM responses by answering and answering verification questions from the model itself is valuable.
3) The proposed method is effective in alleviating hallucination problems.

**Weaknesses:**

1) There is an absence of comparative analysis with other methods aimed at mitigating the hallucination issue. It remains to be clarified whether CoVe offers an enhancement in performance relative to other methods.
2) More instances of prompts are required. For example, in Section 3.3, some examples of prompts need to be provided to distinguish between the several variants of verification variants.
3) Needs to provide more examples of the use of CoVe in different tasks.

**Questions:**

1) How do we verify that the model can do plan verification, execution verification, and verified response well after a few shots? What happens if you use zero-shot CoVe?

2) Is it possible to skip the step of generating a baseline response by directly generating questions similar to plan verification based on the query and generating the final response based on the response? Does CoVe have any advantages over this method?

3) More details of prompts are needed in the supplemental materials, or it would be difficult for readers to follow the ideas.

4) What is the time complexity of the proposed method and other competitors?

---

> ### Author Response · Authors · 2023-11-23
>
> Thanks for the review and for finding our draft well-written. We are grateful for your positive comments on the value and efficacy of our approach.
> We have addressed your concerns and answered your questions below
>
> **Concerns:**
>
> 1. We have now added three new state-of-the-art baselines to the paper. Our work deals with hallucinations specifically for longform generations. We have thus updated Table 3 with baselines using SelfCheckGPT (NLI and LLM) and ChatProtect - three state-of-the-art papers that deal with hallucinations in longform generations. We find our approach, CoVe, outperforms both SelfCheckGPT (with NLI and LLM)  and ChatProtect. Please see our common response for more details.
>
> 2. We add our few-shot prompts in the supplementary material.
>
> 3. Our approach is intended for mitigating hallucinations in longform generations, a highly relevant problem. As LLMs are used more and more as replacements to search engines, it is crucial that we show CoVe reduces hallucinations across 4 generation benchmarks.  We are optimistic that CoVe style verifications can be extended to generative models on real applications, which is beyond the scope of this paper.
>
> **Questions**
>
> 1. Like several reasoning tasks, it is hard to measure the performance of the intermediate parts of the chain owing to the absence of labels, although clearly you can measure the end-to-end performance, or hand annotate some data. For the Wikidata task we manually measure the accuracy of the verification questions and we find that the Llama 65B is around 70% accurate at answering factoid questions related to the professions and birth-places of people. We describe this in the paper under Shortform verification questions are more accurately answered than longform queries.
> 2. That’s an interesting approach to try. For some tasks, it is not likely to work as you will not know which questions to ask, e.g. consider list-based questions (e.g. “who are some politicians born in NYC“), the only question you can really ask at the start is the list-based question itself.
> 3. We had added our few-shot prompts in the supplementary material
> 4. We perform a comparative analysis of the inference overhead encountered. Please check the common response and Section 8 in our appendix.

---

> > ### Comment · Reviewer_kMJo · 2023-12-01
> > **I appreciate the responses by the authors**
> >
> > I appreciate the responses by the authors, and they do solve some of my problems. However, the advantages of the proposed method or the novelty need to be further clarified. Therefore, I maintained my score.

---

### Official Review · Reviewer_r3aM · 2023-11-01

**Soundness:** 2 fair
**Presentation:** 3 good
**Contribution:** 2 fair
**Rating:** 3
**Confidence:** 4

**Summary:**

This paper proposes a new prompting method to mitigate LLMs’ hallucination issues, dubbed chain of verification (CoVE). The model first generates an initial output, and then, conditioning on it together with the input, it generates a series of questions targeting the facts stated in the output. The model then answers the verification questions and identifies the factual errors in the initial response, which is then revised to produce the final outputs. Various variants are explored, each answering the verification questions in different manners. CoVE is tested on several question-answering datasets on Wikipedia, including Wikidata and Wiki category list (both are created from templates), MultiSpanQA, as well biography generation. Results show that CoVE reduces hallucination, at a cost of slightly worse helpfulness.

**Strengths:**

- Mitigating hallucinations in LLMs is a timely and important topic
- CoVE is simple and widely applicable
- Various variants are tested out

**Weaknesses:**

- The technical contribution is thin. I had a hard time justifying the paper’s technical contribution since CoVE looks very similar to [1]
The two Wikipedia QA datasets (Wikidata and Wiki-category list) are created from simple templates and are rather toyish. I am not sure how much they add to the paper.
- Hallucinations are especially tricky to address in generation; the only generation task considered is biography generation, which is way less challenging than most real-world applications
- The paper aims to address hallucination problems. However, I do not find this reflected in the designs of the experiments: 3 out of 4 experiments are question answering, and the remaining one is a rather confined biography generation task. Therefore I find the experiments of this paper weak. Including a more diverse and challenging set of tasks can strengthen the results.
- Wikipedia is a domain that most models have a lot of exposure to; it would be more interesting to see how CoVE performs in other domains such as scientific, legal, and medical domains.
- It seems that CoVE can potentially negatively impact the model’s helpfulness: in Table 1, the number of factual outputs reduces; similarly, in Table 3, CoVE produces fewer facts. Some discussion on this would be interesting.
- I am concerned about making strong conclusions solely based on the FactScore metric: will a model receive higher FactScore by producing shorter outputs with less information? Complementing it with models’ helpfulness, generation quality, and controlling for the number of facts might be necessary

[1] https://arxiv.org/abs/2210.03350

**Questions:**

- A key assumption behind CoVE is that even when the model is not able to generate factual outputs, it might still be able to identify nonfactual statements. A similar observation is mentioned by [2], but they argue that retrieval is necessary. Can the authors elaborate their views on whether or not the model needs external information to identify factual errors in its own outputs?

[2] https://www.semanticscholar.org/paper/Language-Models-Hallucinate%2C-but-May-Excel-at-Fact-Guan-Dodge/45653ad43124f02dc2cf2db3357be1d1d78ddb18?utm_content=title&utm_medium=unfurl&utm_source=slackbot

---

> ### Author Response · Authors · 2023-11-23
>
> We thank you for your review and are appreciative of your positive comments on the applicability of our approach and our experimentation!
>
> We have addressed your concerns below
>
> **Technical contribution**
>
> The paper “Measuring and Narrowing the Compositionality Gap in Language Models” (which is cited) as the title suggests concentrates on compositional questions with multiple hops such as “Who won the Master’s Tournament the year Justin Bieber was born?” – and breaking into subquestions. It does not focus on hallucinations (does not mention that word at all). It uses a method called self-ask to build this subquestion decomposition. In our work questions are used for a different purpose: to verify hallucinations, rather than decompose a multi-hop question, so this is different. Further, we develop joint, 2-step, factored and factor+revise approaches and compare these and show how they are important – none of which is related to this paper.
>
> **Wikidata + Wiki-Category datasets**
>
> We would like to clarify that we do not create the Wiki-Category benchmark ourselves but use a subset of an existing benchmark: Quest (https://aclanthology.org/2023.acl-long.784.pdf).
> Wikidata uses real data from Wikipedia, but the queries are synthetically constructed.
> We also report results on the MultiSpanQA task which does use real queries from humans (from Google). Overall, list-form answers are very natural in assistant-based applications. List-generation questions can be excellent ways to understand LLM hallucinations and rapidly test approaches. List generations mimic longform generations with the advantage that they can be automatically, efficiently and perfectly evaluated, while in other tasks evaluation is much more tricky.
> We agree that questions of the form “Name some politicians born in Boston” appear deceptively trivial **but large-scale commercially deployed LLM systems fail in producing correct answers**. We have provided several qualitative examples of this in the supplementary material.
>
>
>
> **Set of tasks used**
>
> Hallucinations mitigation in longform generations has very recently started receiving significant attention and a lot of benchmarks don’t yet exist for this problem. We chose these four tasks because (i) they are known to produce hallucinations for state-of-the-art models (as shown in experiments), both open source and commercial products; and (ii) ways of evaluating these tasks exist. In particular, the FactScore metric was recently built for the biography task, and hence several other papers, other than ours, are biographies as well (see new baselines).
> We agree that it would be interesting to see how CoVe would perform on further benchmark tasks as they are developed. Scientific, legal and medical domains are great applications and will warrant deep analysis beyond a methods paper such as this one. Hence, this is exciting future work beyond this paper.
>
> **Number of facts vs. number of hallucinations**
>
> Any method that reduces hallucinations will reduce the number of facts. As we note in our research questions, we are interested in whether CoVe can selectively remove fictitious facts with a minimal impact on the number of truthful facts.
> The FactScore metric is normalized by number of facts. So a shorter generation does not automatically mean fewer hallucinations. Note that we also built a baseline whereby the number of facts is reduced by chopping off the ends of passages to match the same number of facts as our best model and we see CoVe significantly outperforms this baseline. We describe this in our passage: CoVe outperforms existing hallucination mitigation baselines.
>
> **Questions**
>
> Our work along with several prior works, eg SelfCheckGPT, ChatProtect have shown that it is possible to mitigate hallucinations without retrieval. We agree that it is impossible to guarantee veracity without access to a verified source of information. We would argue that even retrieval as shown by several  Retrieval Augmented Generation (RAG) style approaches still cannot guarantee this without assuming that everything on the internet is true. We also would like to note that errors can still creep into the RAG pipeline for example the PerplexityAI result from our paper which uses a retrieval system.

---

### Official Review · Reviewer_Lhor · 2023-11-10

**Soundness:** 3 good
**Presentation:** 3 good
**Contribution:** 3 good
**Rating:** 5
**Confidence:** 4

**Summary:**

This paper tackles a hallucination problem of LLMs via a carefully designed framework of iterative prompting. Specifically, the authors propose chain-of-verification (CoVe), which revised the original response of LLMs by generating verification questions and then answering to them. All these processes are conducted via few-shot prompting. Through the experiments on various applications, the effectiveness of CoVe has been demonstrated as it successfully reduces a hallucination of applied LLM (LLaMA-65B) and outperforms the performance of carefully fine-tuned LLM to follow the instruction (LLaMA2-70B Chat).

**Strengths:**

1. **Clarity**. Overall, the writing is clear and easy to follow. In addition, the organization of the main draft is well-established.
2. **Well motivated problem**. Reducing the hallucination and improving the factuality of LLMs is an interesting and important problem. To this end, considering the improved prompting framework is a reasonable and well-motivated direction.
3. **Simple and efficient method.** The proposed method is simple and can be applicable regardless of the types of LLMs. Also, it shows consistent improvement across the various applications.

**Weaknesses:**

1. **Absence of necessary baselines**. As the authors pointed out in the Related work sections, there are many relevant works based on prompting, to reduce the hallucination of LLMs [1,2,3] or improve LLMs’ reasoning [4,5]. However, these baselines are never compared through the experiments now. Therefore, it’s hard to verify the effectiveness of the proposed CoVe compared to them.
2. **Difficulty of direct comparison**. Currently, only zero-shot (or CoT) results are presented for LLaMA2 and few-shot results for LLaMA65B, respectively. It makes be hard to compare both models as there is no overlap. To ease their comparison, including the results of LLaMA2 few-shot and LLaMA zero-shot is strongly encouraged.
3. **Inconsistency across Tables**. While Factor+revise is presented in Table 3 and it achieves the best score, there are no such results in Tables 1 and 2. Does this method not perform well on the setups in Tables 1 and 2? To facilitate the understanding of the working mechanism of the proposed framework.
4. **More qualitative examples**. While the authors present some examples in Figures 1 and 3, it would be better to present more examples to help the understand of readers.

[1] Manakul et al., Selfcheckgpt: Zero-resource black-box hallucination detection for generative large language models., arXiv:2303
[2] Cohen et al., Lm vs lm: Detecting factual errors via cross examination., arXiv:2305
[3] Varshney et al., A stitch in time saves nine: Detecting and mitigating hallucinations of llms by validating low-confidence generation., arXiv:2307
[4] Miao et al., Selfcheck: Using llms to zero-shot check their own step-by-step reasoning., arXIv:2308
[5] Madaan et al., Self-Refine: Iterative Refinement with Self-Feedback., NeurIPS 23

**Questions:**

Please address the concerns in above.

---

> ### Author Response · Authors · 2023-11-23
>
> Thank you for your feedback and we are happy you found our draft clear and easy to follow.
> We agree that hallucinations are an important and interesting problem and we are glad you found our approach well-motivated.
>
> We have responded to your comments below
>
> **Absence of necessary baselines:**
>
> We have added three new state-of-the-art baselines to the paper. Our work deals with hallucinations specifically for longform generations. We have thus updated Table 3 with baselines using SelfCheckGPT (NLI and LLM) and ChatProtect - three state-of-the-art papers that deal with hallucinations in longform generations. We find our approach, CoVe, outperforms both SelfCheckGPT (with NLI and LLM)  and ChatProtect.
>
> **Difficulty of direct comparison**
>
> Llama65B is not an instruction-tuned model and hence produces gibberish (without stopping) when prompted zero-shot. This is expected behavior for a base language model, hence we do not report it (we will clarify this in the paper). When prompted zero-shot with “Let’s think step-by-step”, the model generated erroneous steps, for example, to check information on the web instead of generating the required answer, and did not produce answers.
> We have updated our appendix on the unsuccessful experiments of getting Llama 65B to generate the desired response when prompted zero-shot. In any case, these serve as baselines to CoVe rather than being interesting in themselves to compare. The CoVe experiments are also few-shot using Llama65B, hence the baseline few-shot Llama65B comparison. Llama 2 (which should be superior) is there as a baseline just to show that zero-shot instruction-tuned model baselines do not actually fare better.
>
> **Inconsistency across Tables**
>
> The Factor+revise method adds an additional step where the LLM can check for inconsistencies between the generated content and the answered verification question.
>
> Consider the example
>
>           “Trump received a BS in Business from the University of Pennsylvania”
>            And the verification question and output:
>
>           What degree did Donald Trump graduate with from the University of Pennsylvania in 1968?
>           Answer: A BS in Economics.
>
> The Factor+revise method adds a separate step that has the LLM check if these two outputs are consistent, inconsistent or partially consistent and then recomposes the output based on the consistent parts.
>
> For the set QA tasks checking for inconsistencies and removing an answer from the list is more trivial, as it is just an entity match.
>
> Example:
>
>          Name some politicians born in New York, NY
>         A: Hillary Clinton
>
>         Where was Hillary Clinton born?
>         A: Chicago (Chicago != New York)
>
> CoVe - 2 step is a variant of CoVe-Factored where the Verification questions are evaluated in one step (thus reducing the need to prompt the LLM several times for each verification question). This is feasible to use on tasks such as set QA but as the number of verification questions grows answering them all can be difficult in the fixed context size.
> We hence don’t experiment with this approach on the harder benchmarks.
>
>
> **More qualitative examples**
>
> Figure 1 and 3 show full examples of CoVe.
> There is also analysis in section 9 of how and why ChatGPT fails on a number of tasks, and why verification questions can work.
> Finally, we have now added the prompts used as part of our supplementary material

---

> > ### Comment · Reviewer_Lhor · 2023-12-01
> > **Response to Authors**
> >
> > I appreciate the effort that the authors put into addressing my questions. I believe that the above results and discussion can significantly improve the quality of the manuscript. However, the draft still has room for improvement and it seems to be better to improve and consider the other venues. Therefore, I will keep my original rating.

---

### Author Response · Authors · 2023-11-23
**Common Response**

We thank all the reviewers for their feedback which has only helped improve our work.
Based on the reviewers' feedback, we have added and compared our approach to existing methods.
We also are grateful to the reviewers for encouraging us to think about the efficiency of our approach.
We perform an analysis comparing the inference efficiency of our approach against other hallucination mitigation methods.

**CoVe**

We proposed a simple but effective approach to mitigate hallucinations in longform text.
We evaluate the efficacy of our approach on several longform benchmarks and show that it can reduce hallucination more effectively than other recently proposed methods.


**Baselines**

We have added 3 more baselines. We compare CoVe to 3 recent state-of-the-art approaches for longform hallucination mitigation – the NLI and LLM variants of SelfCheckGPT (the best-performing variants) and ChatProtect.
We have added our results to our draft in Table 3.
CoVe performs well compared to these methods.

| Base Models        | Method                 | FACTSCORE  | Avg # facts |
|--------------------|------------------------|------------|-------------|
| GPT3               | Few-shot               | 45.3       | 15.6        |
| GPT3 + ChatGPT     | ChatProtect            | 48.5       | 14.6        |
| GPT3 + DeBERTA     | SelfCheckGPT-NLI       | 60.6       | 6.0         |
| GPT3 + InstructGPT | SelfCheckGPT-LLM       | 61.7       | 6.3         |
| GPT3 + InstructGPT | CoVe - Factor + revise | **68.6**       | 9.0         |




**Inference overhead**

CoVe essentially requires 4 types of LLM prompt calls, corresponding to the 4 steps of the method (1. Generate baseline, 2. Plan verifications, 3. Execute verifications, 4. Final response). Step 3 in the factored variant of the approach is actually several LLM calls, one for each planned verification question, however, they can all be called in parallel and hence use batching.

We have computed the approximate number of LLM calls CoVe would need to perform.
We also perform a comparative analysis for LLM calls required for other existing hallucination mitigation approaches.
We find that CoVe has a comparable inference overhead to other approaches. We describe our analysis further in Section 8 in our appendix.


| Base Models        | Method                 | Worst case LLM calls                      |
|--------------------|------------------------|-------------------------------------------|
| GPT3               | Few-shot               | 1                                         |
| GPT3 + ChatGPT     | ChatProtect            | 1 + num_sents x repeats x 3               |
| GPT3 + InstructGPT | SelfCheckGPT-LLM       | num_samples + 1 + num_sents x num_samples |
| GPT3 + InstructGPT | CoVe - Factor + revise | 1 + num_sents + 2 x num_facts             |


**Prompts**

We have now included the Prompts we use for each step of CoVe in our supplementary material for reproducibility.

---

### Meta-Review · Area_Chair_kBNJ · 2023-12-06

**Metareview:**

The paper is well-motivated, presenting a simple and efficient approach to tackle hallucination by independently answering fact-check questions. Reviewers raised questions about baselines, direct comparisons, technical contributions, results in other domains, and variations in the few-shot examples used. I appreciate the rebuttal and additional comments provided by the authors. However, the impact of the technical contribution remains unclear. For instance, the proposed approach has already been utilized in [https://arxiv.org/abs/2210.03350], though the target tasks differ. Considering the overall ratings from the reviewers and the assessment of the rebuttal and additional comments, I recommend rejecting this submission

**Justification For Why Not Higher Score:**

As mentioned above, the impact of the technical contribution remains unclear after reading the rebuttal and additional comments.

**Justification For Why Not Lower Score:**

N/A

---

### Decision · Program_Chairs · 2024-01-16

Reject